# FLOW MATCHING GENERALIZES THROUGH DISCRETIZATION BIAS

## ABSTRACT

Flow models exhibit an extraordinary ability to generalize, generating realistic samples far beyond the training data. This phenomenon lacks a simple explanation. We argue that the key mechanism is not the accurate solution of a continuous-time ODE, but rather the error introduced by its discretization. To isolate this effect within the flow matching framework, we introduce the *Empirical Velocity Field (EVF)*, a non-parametric estimator of the *conditional velocity field* derived by replacing the target distribution with its empirical measure. The exact ODE flow driven by the EVF turns out to be uninteresting, yielding a kernel density estimate that collapses onto the training data. However, its discretization is remarkably powerful. We show that even a single Euler step induces a projection-like effect, concentrating samples on the underlying data manifold and creating diverse, high-quality samples. We support this with extensive empirical evidence and provide a theoretical analysis of the one-step estimator that quantifies this projection, offering a rigorous foundation for how discretization generates structured samples. Our findings argue that the generative success of flow matching is fundamentally driven by the implicit bias of numerical ODE solvers.

## 1 INTRODUCTION

Flow-based generative models exhibit an extraordinary ability to generalize, creating realistic and diverse samples that appear to be drawn from the same underlying distribution as the training data, yet are entirely novel. This phenomenon, central to their success in domains like image synthesis, lacks a simple, satisfying explanation. While theories based on statistical minimax optimality establish their prowess in density estimation (Gao et al., 2024), they fail to explain why flow models dramatically outperform other optimal estimators like Kernel Density Estimates (KDE) (Tsybakov, 2008). Sampling from a KDE, after all, amounts to selecting a training example and adding noise—a procedure that fails to produce the rich, structured novelty characteristic of modern generative models. What, then, is the fundamental mechanism driving their generative power?

In this paper, we argue that the key to this generalization is not the accurate solution of a continuous-time Ordinary Differential Equation (ODE), but rather the *implicit bias* introduced by its numerical discretization. This is a deeply counter-intuitive claim: discretization error is typically viewed as a nuisance to be minimized, not a feature to be embraced. We argue, however, that this "error" is precisely what transforms a simple interpolative procedure into a powerful generative one.

To isolate and study this effect, we strip away the complexities of neural network approximation and introduce the *Empirical Velocity Field (EVF)*. The EVF is a non-parametric estimator of the conditional velocity field, derived by replacing the unknown target distribution with the empirical measure of the training data. This provides a closed-form expression for the velocity field, allowing us to cleanly separate the properties of the flow itself from the effects of numerical integration.

The EVF serves as a powerful analytical tool. When we consider the exact, continuous-time ODE flow driven by the EVF, the outcome is uninteresting for generation: it yields a variant of a KDE that, as time $t \to 1$, simply collapses onto the training samples. This confirms that the underlying continuous dynamics offer no mechanism for creating novel samples beyond the training set.

The magic happens upon discretization. We show that even a single step, at time $t$ near 1, of a numerical solver, like the Euler method, dramatically alters the outcome. This single discretized step

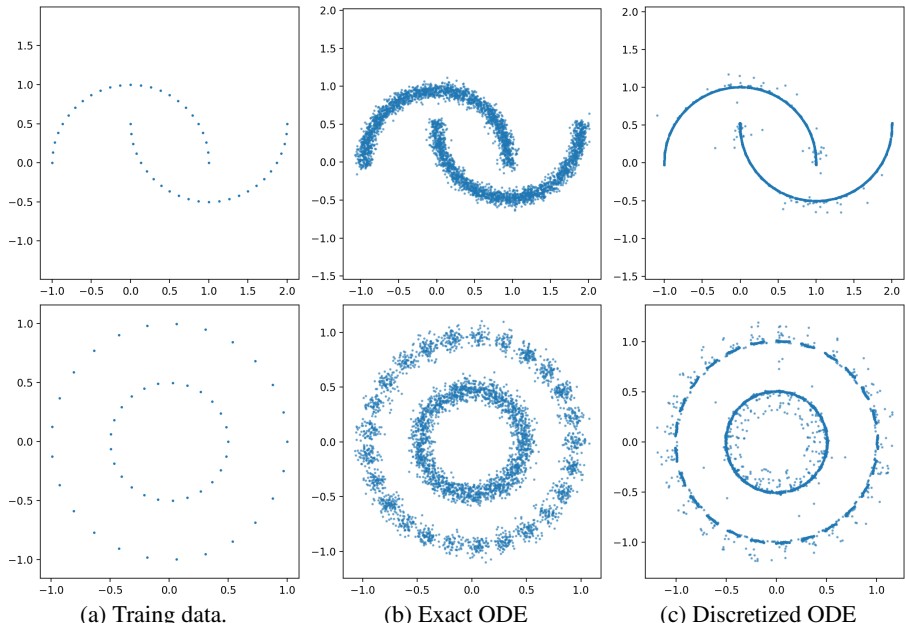

(a) Traing data.  (b) Exact ODE  (c) Discretized ODE

Figure 1: Two moons and two circles data. (a) $n = 50$ samples from the target distribution used for training. (b) The exact ODE solution for EVF at $t = 0.95$. (c) The discretized ODE solution, 10 steps, using 2nd order Runge–Kutta method, with final field evaluation at the same value of $t = 0.95$. For each of cases (b) and (c) 4096 new samples are generated. See Section 3.1 for more details.

induces a powerful projection-like effect, taking points from a diffuse, off-manifold distribution and concentrating them onto or near the underlying data manifold. As illustrated in Figure 1, this process fills gaps between training examples, generating diverse, high-quality samples that respect the intrinsic structure of the data. We provide a theoretical analysis of this one-step generator that quantifies this projection effect, offering a rigorous foundation for how discretization error manufactures structured novelty.

Our contributions are three fold:

1. We introduce the Empirical Velocity Field (EVF) as a powerful tool for analyzing flow matching and as a simple, effective method for unconditional generation.

2. We present the central thesis that the generative power of flow models stems from the implicit bias of numerical ODE solvers, contrasting the generative failure of the exact ODE flow (a KDE) with the success of its discretized counterpart.

3. We support this thesis with extensive empirical evidence on toy and image datasets, using standard and novel evaluation metrics, and provide a theoretical analysis that formalizes the projection-like behavior of the discretization.

Our findings suggest a paradigm shift in understanding and developing flow-based models. The goal may not be to find ever-more-accurate ODE solvers, but to design numerical integration schemes whose biases are intentionally structured to promote generalization and sample quality.

## 1.1 RELATED WORK

Flow-based neural ODE generators trained via simulation-free regression have evolved along related lines. Flow matching Lipman et al. (2023) learns a time-dependent velocity field by supervised regression to analytically defined targets along simple interpolation paths in data space. Training and sampling can be moved to the latent space of a pretrained autoencoder Dao et al. (2023) for efficiency and scalability. A recurring aim is to 'straighten' probability paths to reduce curvature and stiffness (Kornilov et al., 2024), simplifying training and accelerating ODE sampling: multi-sample couplings straighten flows and improve sample efficiency Pooladian et al. (2023); rectified

flow Liu et al. (2023) makes trajectories near-linear for faster, more stable training and sampling; and large transformer backbones have been scaled for rectified-flow models to achieve state-of-the-art high-resolution synthesis with few function evaluations Esser et al. (2024). Coupling and conditioning have also advanced the field: minibatch optimal transport improves source–target pairings, stabilizing training and inference Tong et al. (2023a); and conditional flow matching (CFM) learns condition-dependent velocity fields for flexible conditional generation without stochastic simulation or likelihood-based training Tong et al. (2023b). Our focus is orthogonal: we study how discretization improves image generation quality and explain why flow-matching methods work well, rather than competing with these approaches.

This work studies generation by interpolation through the lens of stochastic interpolants Albergo et al. (2023), which provide closed-form expressions for time-dependent distributions and velocity fields induced by chosen interpolations, unifying score-based diffusion, probability flow ODEs, and flow-matching-style training. While the analysis in Albergo et al. (2023) focuses on settings where the target distribution $\rho_1$ is modeled as a Gaussian mixture, our setting considers an empirical distribution of $Y$ in $\rho_1$. Moreover, the proposed empirical velocity field (EVF) directly uses the closed-form velocity as the training target and also as the deployed velocity field, which, to our knowledge, was not explored in Albergo et al. (2023).

Our work relates to the manifold learning literature, which has been extensively studied in machine learning and statistics Ma & Fu (2012). Our setting is simpler than denoising observations from an unknown manifold. We observe that, for $x_t$ near the manifold, Euler's method behaves like a weighted average of nearby samples, effectively projecting $x_t$ onto the manifold. This differs from the local PCA approach of Lin & Zha (2008), yet still recovers samples close to the manifold.

## 2 FLOW MATCHING AND THE EMPIRICAL VELOCITY FIELD

To isolate the effect of discretization, we first need a precise, analytic form for the velocity field, free from the confounding effects of neural network approximation. We achieve this by developing the Empirical Velocity Field (EVF), a non-parametric estimator derived directly from the principles of flow matching.

### 2.1 BACKGROUND: CONDITIONAL FLOW MATCHING

Flow models aim to transport samples from a simple prior distribution $P_0$ (e.g., a standard Gaussian $Z \sim N(0, I_d)$) to a complex target distribution $P_1$ (e.g., of data $Y$) by integrating an Ordinary Differential Equation (ODE):

$$\frac{dX_t}{dt} = v(t, X_t), \quad X_0 = Z, \tag{1}$$

where $v(t, x)$ is a time-dependent velocity field. The core idea of flow matching Lipman et al. (2023) is to define a probability path $P_t$ that interpolates between $P_0$ and $P_1$, and then find the unique velocity field $v(t, x)$ that induces this path.

A common choice is the linear interpolation path Albergo & Vanden-Eijnden (2023), where the random variable $X_t$ at time $t$ has the same distribution as $I_t(X_0, X_1) := (1 - t)X_0 + tX_1$. The corresponding velocity field is given by the conditional expectation of the path's velocity:

$$v(t, x) := \mathbb{E}\big[\partial_t I_t(X_0, X_1) \,|\, X_t = x\big] = \mathbb{E}[X_1 - X_0 \,|\, X_t = x]. \tag{2}$$

In practice, $v(t, x)$ is approximated by a neural network $v_\theta(t, x)$, which is trained via regression by minimizing a loss like $\mathbb{E}_{t, X_0, X_1}[\|v_\theta(t, X_t) - (X_1 - X_0)\|^2]$. This training process, however, is noisy and its regularizing effects are intertwined with those of the subsequent ODE solve.

### 2.2 THE EMPIRICAL VELOCITY FIELD: A NON-PARAMETRIC ALTERNATIVE

To disentangle these effects, we consider a direct, non-parametric estimator for the velocity field. First, we rewrite the conditional velocity field in a more convenient form:

$$v(t, x) = \frac{1}{1 - t}\Big(\mathbb{E}[X_1 | X_t = x] - x\Big), \tag{3}$$

which follows from the linearity of expectation and the relation $X_0 = (X_t - tX_1)/(1-t)$.

Now, instead of learning a parametric approximation, we construct the *Empirical Velocity Field (EVF)* by replacing the true target distribution $P_1$ with the empirical distribution of the training data $\{y_1, \ldots, y_n\}$, denoted $\mathbb{P}_n := n^{-1} \sum_{i=1}^n \delta_{y_i}$. This plug-in approach yields a closed-form expression for the velocity field.

**Proposition 1.** *Let the prior $P_0$ be a distribution with density $f_Z$, and the target $P_1$ be the empirical distribution $\mathbb{P}_n$. For the linear interpolation path, the velocity field $v_{EVF}(t, x)$ and the density $\rho_{EVF}(t, x)$ of the exact ODE solution $X_t$ are given by:*

$$v_{EVF}(t, x) = \frac{1}{1-t} \left( \frac{\sum_{i=1}^n y_i f_Z(\frac{x-ty_i}{1-t})}{\sum_{i=1}^n f_Z(\frac{x-ty_i}{1-t})} - x \right), \tag{4}$$

$$\rho_{EVF}(t, x) = \frac{1}{n} \sum_{i=1}^n \frac{1}{(1-t)^D} f_Z \left( \frac{x - ty_i}{1-t} \right). \tag{5}$$

The proof is provided in the Appendix B. The expression for $\rho_{\text{EVF}}(t, x)$ is particularly revealing. It is precisely the density of a Kernel Density Estimate (KDE) of the scaled data $\{ty_i\}_{i=1}^n$, using the prior density $f_Z$ as a kernel with bandwidth $h = 1 - t$.

This leads to a crucial insight: as $t \to 1$, the bandwidth $h \to 0$, and the distribution of the exact ODE solution $X_t$ converges to the empirical distribution $\mathbb{P}_n$. Consequently, **solving the continuous-time ODE driven by the EVF is generatively uninteresting**. It is equivalent to sampling from a KDE that sharpens to eventually just return the training samples themselves. It provides no mechanism for creating novel samples that intelligently fill the gaps between training data. The remarkable generative power of flow matching must, therefore, originate from another source.

## 2.3 EVF AS A STRONG VELOCITY FIELD ESTIMATOR

Before proceeding, we briefly establish that the EVF is not merely a theoretical construct, but a powerful and sample-efficient estimator in its own right. We compare the generative performance of an ODE solver driven by our EVF against one driven by a standard Neural Network Velocity Field (NNVF), trained on the same data. Figure 2 shows this comparison.

For the NNVF, we adopt the architecture and training setup from the first example in Lipman et al. (2024). The network is a multi-layer perceptron (MLP) with three hidden layers of 64 units each and the ELU activation function. We train it for 10,000 steps using the Adam optimizer with a learning rate of $10^{-2}$ and a batch size of 256. While the original example samples fresh data for each batch (hence seeing a total of $10000 \times 256$ training samples), we constrain the training to a fixed dataset of $n = 1024$ samples to simulate a realistic, data-limited setting. Both the EVF and the trained NNVF are then used to generate samples via a discretized ODE solver (specifically, a 2nd-order Runge-Kutta method with 10 steps).

The results in Figure 2 are striking. The discretized ODE using the EVF generates samples that are much more clearly structured and faithful to the underlying manifold than those generated using the trained NNVF. This superior sample efficiency validates the EVF as a high-quality field estimator and confirms that by using it, we are not losing fidelity compared to the standard neural network approach. Its strength allows us to confidently use it as a tool to isolate the role of the numerical solver. Having established that the exact, continuous-time EVF flow fails to generalize, we now investigate the impact of its discretization.

## 3 DISCRETIZATION BIAS AS A GENERATIVE MECHANISM

We have established that the exact, continuous-time flow driven by the EVF is equivalent to a Kernel Density Estimate (KDE), which fails to produce novel, structured samples. We now demonstrate that the numerical discretization of this very same ODE is a powerful generative process. The "error" introduced by the solver is, in fact, the source of generalization.

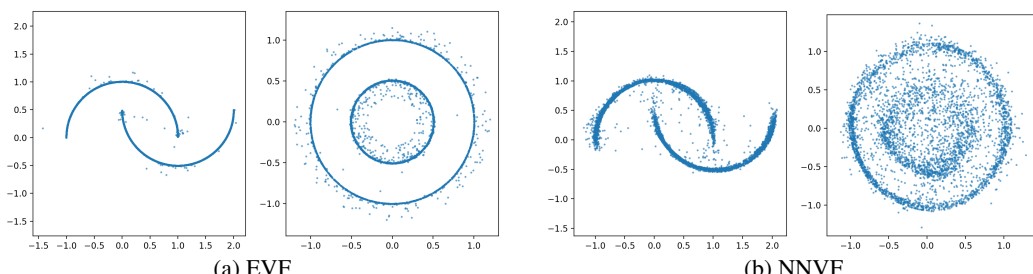

Figure 2: Comparison of discretized ODE (D-ODE) samples generated using the Empirical Velocity Field (EVF) versus a trained Neural Network Velocity Field (NNVF). Both are based on the same $n = 1024$ training samples. The EVF produces samples that are more tightly concentrated on the true data manifolds.

### 3.1 The Stark Contrast: Exact vs. Discretized Flows

The difference between the exact and discretized solutions is not subtle; it is a dramatic qualitative shift. Figure 1 provides a clear visualization of this phenomenon. We start with a sparse training set of $n = 50$ samples from two synthetic manifolds.

**The Exact ODE Solution** (Fig. 1b), corresponding to sampling from the density $\rho_{\text{EVF}}(t, x)$ at $t = 0.95$, behaves exactly as our theory predicts. The samples form diffuse clouds around the original training points. This is classic KDE behavior: it smoothes the empirical measure but does not "understand" the underlying manifold structure to fill in the large gaps between training points.

**The Discretized ODE Solution** (Fig. 1c), generated by integrating the EVF with a standard 10-step numerical solver, is completely different. The samples are sharply concentrated on the true data manifold, perfectly interpolating the gaps in the training data. This process generates entirely new points that are nevertheless highly plausible, demonstrating true generalization.

This stark contrast makes our central hypothesis tangible: the discretization scheme is not merely approximating the continuous flow; it is introducing a strong inductive bias that favors on-manifold solutions.

### 3.2 Analysis of the One-Step Generator: A Projection Effect

To understand how discretization achieves this, we can analyze its effect in the simplest non-trivial case: a single Euler step from a time $t$ very close to 1, to the final time $t = 1$. Let $x_t$ be a point sampled from the exact KDE-like distribution $\rho_{\text{EVF}}(t, x)$. The one-step Euler update is:

$$x_1 = x_t + (1 - t) \cdot v_{\text{EVF}}(t, x_t) \tag{6}$$

$$= x_t + (1 - t) \cdot \frac{1}{1 - t} \left( \frac{\sum_{i=1}^{n} y_i f_Z(\frac{x_t - t y_i}{1 - t})}{\sum_{i=1}^{n} f_Z(\frac{x_t - t y_i}{1 - t})} - x_t \right) \tag{7}$$

$$= \frac{\sum_{i=1}^{n} y_i f_Z(\frac{x_t - t y_i}{1 - t})}{\sum_{i=1}^{n} f_Z(\frac{x_t - t y_i}{1 - t})}. \tag{8}$$

This final expression is a Nadaraya-Watson kernel regression estimator. It estimates the value of a function at $x_t$ by taking a weighted average of the "target values" $y_i$. The weights are determined by the kernel $f_Z$, centered at each $t y_i$. Intuitively, the estimator pulls the point $x_t$ towards a weighted average of the nearby training samples $\{y_i\}$.

When the data $\{y_i\}$ lie on or near a low-dimensional manifold $\mathcal{M}$, this averaging process has a powerful geometric consequence: it acts like a projection. If $x_t$ is close to a region of the manifold, the kernel $f_Z(\cdot)$ will assign significant weight only to the training points $y_i$ in that local neighborhood. Because the manifold is locally flat, this weighted average of nearby on-manifold points will lie very close to the manifold itself—much closer, in fact, than the original point $x_t$.

We formalize this intuition in the following theorem, which shows that the distance of the generated point to the manifold shrinks quadratically with the step size $h = 1 - t$. For simplicity, the theorem analyzes a slightly modified estimator where the kernel is centered on $y_i$ instead of $ty_i$, which is a very accurate approximation for $t \approx 1$.

**Theorem 1** (Projection Effect). Let the training data $\{y_i\}_{i=1}^n$ lie on a smooth manifold $\mathcal{M}$. Consider the one-step generator

$$\hat{y} = \sum_{i=1}^n w_i y_i, \quad \text{where} \quad w_i = \frac{f_Z((x - y_i)/h)}{\sum_{j=1}^n f_Z((x - y_j)/h)}, \tag{9}$$

for some point $x$ and bandwidth $h$. Assume:

(i) Density $f_Z$ has compact support on a ball of radius $r$, $B_r(0)$.

(ii) The manifold $\mathcal{M}$ is sufficiently smooth: Denoting the affine tangent space at $u$ as $\mathcal{T}_u = u + T_u\mathcal{M}$, there exists $\kappa, R > 0$, such that for any $u, v \in \mathcal{M}$, if $\|u - v\| \leq R$, then $\|v - P_{\mathcal{T}_u}(v)\| \leq \kappa\|P_{T_u\mathcal{M}}(u - v)\|^2$.

Let $\pi(\hat{y})$ be the unique closest-point projection of $\hat{y}$ onto $\mathcal{M}$, and let $h := 1 - t$. If $4rh \leq R$, then the distance of the generated sample to the manifold is bounded by:

$$\|\hat{y} - \pi(\hat{y})\| \leq 4\kappa r^2 h^2.$$

The proof is provided in the Appendix C. This result rigorously shows that the one-step generator projects points towards the manifold. For bandwidth $h = 1 - t$, the sample of $x$ has a average distance $h\sqrt{d}$ by the formula $\rho_{\text{EVF}}$ derived in (5). This theorem shows the discretization can reduce that dramatically to $O(h^2)$. The quadratic dependence on $h$ implies that even for a moderately small step size (i.e., $t$ close to 1), the generated points will be extremely close to $\mathcal{M}$. This explains the sharp, on-manifold samples seen in Figure 1c. The discretization bias is not random noise; it is a structured mechanism that enforces the manifold hypothesis, effectively learning the geometry of the data from the training samples. This single step is so powerful that it motivates a highly efficient generative algorithm, which we call *Euler-1*, consisting of sampling $x_t$ and applying a single update.

While Theorem 1 establishes the *fidelity* of the generated samples (they are close to the manifold), it does not guarantee *diversity* (they cover the manifold). A second result confirms that our generator can indeed produce a rich variety of samples. It shows that any point $u$ on the manifold can be generated, provided it is "reachable" by the generator from some point $x$ in the input space. Since the input distribution $\rho_{\text{EVF}}(t, x)$ is diffuse, it is highly likely that such points $x$ will be sampled, leading to broad coverage of the manifold. We state this formally as Theorem 2 (see Appendix D for proof), which ensures that the support of the generated distribution is not arbitrarily constrained.

**Theorem 2** (Diversity). We assume $f_Z$ is continuous. For $u \in \mathcal{M}$, suppose there exists $x \in \mathbb{R}^D$, such that

(i) $x$ is an interior point of the support of its density function.

(ii) $u = \pi(\hat{y}(x))$ with $\hat{y}(x) = \hat{y}$ in (9).

then the random variable $\pi(\hat{y})$ has positive density value at $u$ for a continuous density function.

Assumption (ii) is easy to satisfy in practice: $u$ lies on a $d$-dimensional manifold, while $x$ can be chosen in the higher-dimensional ambient space $\mathbb{R}^D$, offering enough degrees of freedom to ensure $u = \pi(\hat{y}(x))$ for at least one $x$.

## 4 EXPERIMENTAL FRAMEWORK

To empirically validate our thesis, we need metrics that can distinguish between mere memorization and true generalization. While standard precision and recall are useful, they can be misleadingly high for a model that simply reproduces the training data. We therefore introduce a novel, conditioned version of these metrics designed specifically to measure the ability to generate high-quality samples *away* from the training set.

### 4.1 Standard Precision and Recall

We first build on the non-parametric precision and recall metrics of Kynkäänniemi et al. (2019). Given a set of real samples $S_{\text{real}} \sim P_{\text{real}}$ and a set of generated samples $S_{\text{gen}} \sim P_{\text{gen}}$, we estimate the support of their respective distributions by constructing manifolds, $\widehat{\mathcal{M}}_{\text{real}}$ and $\widehat{\mathcal{M}}_{\text{gen}}$. Specifically, for any set $S = \{x_i\}$, the manifold estimate is the union of hyperspheres around each point: $\widehat{\mathcal{M}}(S) = \bigcup_{x_i \in S} B(x_i, r_i)$, where the radius $r_i$ is the distance to the $k$-th nearest neighbor of $x_i$ within $S$. Following common practice, we use $k = 3$.

Precision and recall are then defined as the fraction of samples from one distribution that fall within the estimated manifold of the other:

$$\text{Precision} = P(X_{\text{gen}} \in \widehat{\mathcal{M}}_{\text{real}}) \quad \text{Recall} = P(X_{\text{real}} \in \widehat{\mathcal{M}}_{\text{gen}}) \tag{10}$$

Precision measures fidelity (are generated samples realistic?), while recall measures diversity (does the generator cover the full variety of real data?).

### 4.2 Novelty-Conditioned Precision and Recall (NcPR)

A key limitation of the standard metrics is that a generator that only memorizes the training set, $S_{\text{train}}$, can achieve (near) perfect precision and recall, when training set is large enough, without demonstrating true generalization. To specifically assess a model's ability to generalize, we introduce *Novelty-Conditioned Precision and Recall (NcPR)*.

The core idea is to restrict the evaluation to samples that are demonstrably "novel" with respect to the training set. We quantify novelty by the Euclidean distance to the nearest training sample: $d(x, S_{\text{train}}) = \min_{z \in S_{\text{train}}} \|x - z\|_2$. We then filter the real and generated sample sets to retain only those points that are furthest from the training data.

Let $S_{\text{gen}}(p_g)$ be the subset of $S_{\text{gen}}$ containing the $(1-p_g)$ fraction of samples with the largest distance to $S_{\text{train}}$ (i.e., the top $(1-p_g)$ quantile of novelty). Similarly, let $S_{\text{real}}(p_r)$ be the corresponding novel subset of real data. We then construct manifolds using only these novel subsets:

$$\widehat{\mathcal{M}}_{\text{gen}}(p_g) = \widehat{\mathcal{M}}(S_{\text{gen}}(p_g)) \quad \text{and} \quad \widehat{\mathcal{M}}_{\text{real}}(p_r) = \widehat{\mathcal{M}}(S_{\text{real}}(p_r)). \tag{11}$$

The NcPR metrics are then defined by evaluating precision and recall on these conditioned sets:

$$\text{NcP}(p_g, p_r) = P(X_{\text{gen}} \in \widehat{\mathcal{M}}_{\text{real}}(p_r) \mid X_{\text{gen}} \in S_{\text{gen}}(p_g)) \tag{12}$$

$$\text{NcR}(p_g, p_r) = P(X_{\text{real}} \in \widehat{\mathcal{M}}_{\text{gen}}(p_g) \mid X_{\text{real}} \in S_{\text{real}}(p_r)) \tag{13}$$

In practice, we estimate these by computing standard precision and recall but using the filtered sets $S_{\text{gen}}(p_g)$ and $S_{\text{real}}(p_r)$. By setting $p_g > 0$ and $p_r > 0$, we explicitly ask: "Among the most novel generated samples, what fraction are realistic? And do these novel samples cover the most novel real data?" This provides a direct measure of generalization, heavily penalizing models that simply stay close to their training data. Setting $p_g = p_r = 0$ recovers the standard metrics.

**Remark 1.** For image experiments, direct pixel-wise comparison is not meaningful. We follow standard practice and first map all images into a pre-trained feature space (in our case, the pool3 layer of Inception-v3 (Szegedy et al., 2016)) before computing distances and applying the metrics.

## 5 Experiments and Results

We now present empirical results across a range of datasets to demonstrate the generative power of discretization bias. We compare three generators based on the EVF, contrasting the exact solution with its discretized counterparts, and use our Novelty-Conditioned Precision and Recall (NcPR) metric to quantify true generalization.

### 5.1 Experimental Setup

**Datasets.** We evaluate our methods on two synthetic datasets and two standard image benchmarks:

- **Two Moons:** A classic 2D manifold problem, allowing for direct visualization.

- **Variable Circles:** A synthetic $32{\times}32$ grayscale image dataset we introduce, where each image contains a circular ring with a random center and radius. This provides a controllable image manifold to test generalization beyond the specific circles seen during training.
- **MNIST** ($28{\times}28$) Deng (2012) and **CIFAR-10** ($32{\times}32$) Krizhevsky et al. (2009).

For Two Moons, we use a small training set of $n_{\text{train}} = 50$. For all image datasets, we use $n_{\text{train}} = 1024$ samples. Evaluation is performed against a held-out test set of 2048 samples. More experiments, including high-dimensional synthetic data can be found in the appendix.

**Methods Compared.** We evaluate three generators derived from the EVF, plus a baseline:

- **Exact-$x_t$:** Samples are drawn directly from the analytical density $\rho_{\text{EVF}}(t, x)$. This represents the exact ODE solution at time $t$ and is equivalent to a KDE. We vary the time parameter $t \in [0, 1)$.
- **Euler-1:** Our proposed one-step generator. We first sample $x_t \sim \rho_{\text{EVF}}(t, x)$ and then apply a single forward Euler step to evolve the sample to $t = 1$, as described in Section 3.2. We vary the starting time $t$.
- **D-ODE (rk2):** A multi-step discretized ODE solver. We integrate the EVF from $t = 0$ to $t = 1$ using a 2nd-order Runge-Kutta (rk2) method with a varying number of steps, $s$.
- **Train:** A baseline "generator" that simply consists of the training set itself. This helps calibrate our NcPR metric, as it represents a case of pure memorization with zero novelty.

**Metrics.** As defined in Section 4, we report standard Precision/Recall (PR) and Novelty-Conditioned PR (NcPR). For NcPR, we use thresholds of $(p_g, p_r) = (0.95, 0.5)$, meaning we evaluate the precision of the 5% most novel generated samples against the manifold formed by the 50% most novel real samples, and vice versa for recall. This stringent condition on generated samples ($p_g = 0.95$) focuses the evaluation squarely on the model's ability to extrapolate.

## 5.2 RESULTS

Figure 3 summarizes our findings across all datasets. Each curve shows how a method's PR/NcPR trade-off evolves as its main parameter ($t$ or $s$) is varied. The top row shows standard PR, while the bottom row shows NcPR. Real images generated by these methods appear in Appendix A.

**Discretization is the Key Generative Step.** In both standard and novelty-conditioned evaluations, the **Exact-$x_t$** generator (blue curves) consistently exhibits poor performance: Either low recall or low precision, often both.

In sharp contrast, both the **Euler-1** (green) and **D-ODE** (red) generators achieve significantly higher precision and recall, forming tight clusters in the desirable top-right corner of the plots. This establishes that numerical discretization is the critical element that transforms the generatively weak EVF flow into a high-fidelity generative process.

**NcPR Confirms Generalization is the Driving Force.** The NcPR plots (bottom row) provide a deeper insight, revealing that this performance gain is driven by true generalization. By focusing the evaluation on novel samples, NcPR amplifies the performance gap. While the discretized methods maintain their excellent scores, the performance of Exact-$x_t$ remains poor. The discretized methods, however, excel at creating samples that are simultaneously novel and realistic (see also Appendix A).

The **Train** baseline (gray marker) further calibrates this interpretation. In the low-sample regime of Two Moons ($n = 50$), its NcP drops to nearly zero, as expected for a method with no novelty, while both discretized generators achieve a significantly higher NcP. This perfectly captures the intuitive contrast between panels (a) and (c) of Figure 1. On the denser image datasets, **Train** retains a higher NcP, showing that designing a novelty-conditioned metric is harder for larger training sets.

In summary, the empirical results are unequivocal. The continuous-time flow defined by the EVF is a poor generator. However, its numerical discretization, even with a single step, introduces a powerful generative bias that produces diverse, high-quality samples lying on the data manifold, far from the training examples. Discretization is not a bug; it is the engine of generalization.

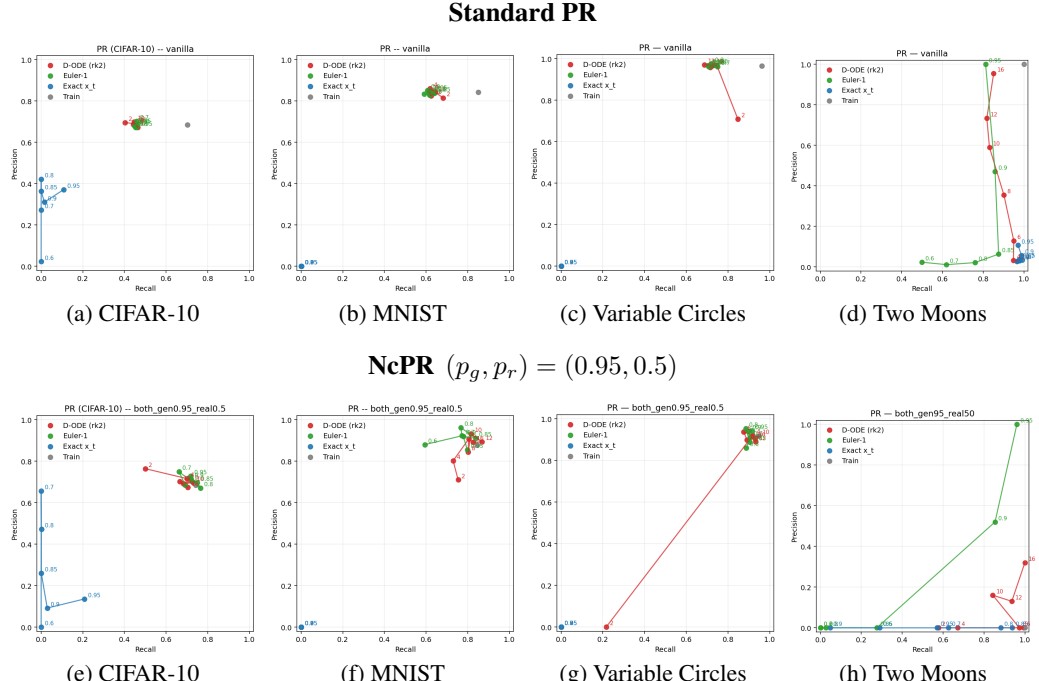

Figure 3: Precision–Recall (PR) comparison across the four datasets. Columns correspond to datasets; the top row shows standard PR, and the bottom row shows Novelty-Conditioned PR (NcPR) with novelty thresholds $(p_g, p_r) = (0.95, 0.5)$ (see Section 4). Each point summarizes a generator/solver setting; gray markers show training baselines. See appendix for larger versions.

## 6 CONCLUSION

The remarkable generalization capability of flow-based models has been a puzzle. In this work, we have argued that the solution lies not in the fidelity of the continuous ODE approximation, but in the implicit bias of the numerical solvers used to integrate it. We isolated this phenomenon by introducing the Empirical Velocity Field (EVF), a non-parametric estimator that allowed us to study the flow dynamics without the confounding effects of neural network training.

Our analysis revealed a stark dichotomy. The exact ODE flow driven by the EVF is generatively powerless, producing samples equivalent to a Kernel Density Estimate that collapses onto the training data. In contrast, its numerical discretization, even with a single Euler step, acts as a powerful generative mechanism. We provided a theoretical analysis showing that this discretization has a projection-like effect, concentrating samples onto the underlying data manifold. This explains how flow models create novel, high-fidelity samples that intelligently fill the gaps in the training set. Extensive experiments, evaluated with our proposed Novelty-Conditioned Precision and Recall (NcPR) metric, provided unequivocal empirical support for this conclusion.

This finding challenges the conventional wisdom that discretization error is something to be minimized. For generative modeling with flows, it appears to be the very source of generalization. This perspective opens up exciting new avenues for future research. Instead of focusing solely on more accurate ODE solvers, perhaps we should be designing and analyzing numerical integration schemes specifically for their generative properties. Understanding and controlling the implicit bias of different solvers could be the key to developing a new generation of more efficient, more powerful, and better-understood generative models.

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

## A  GENERATED IMAGES

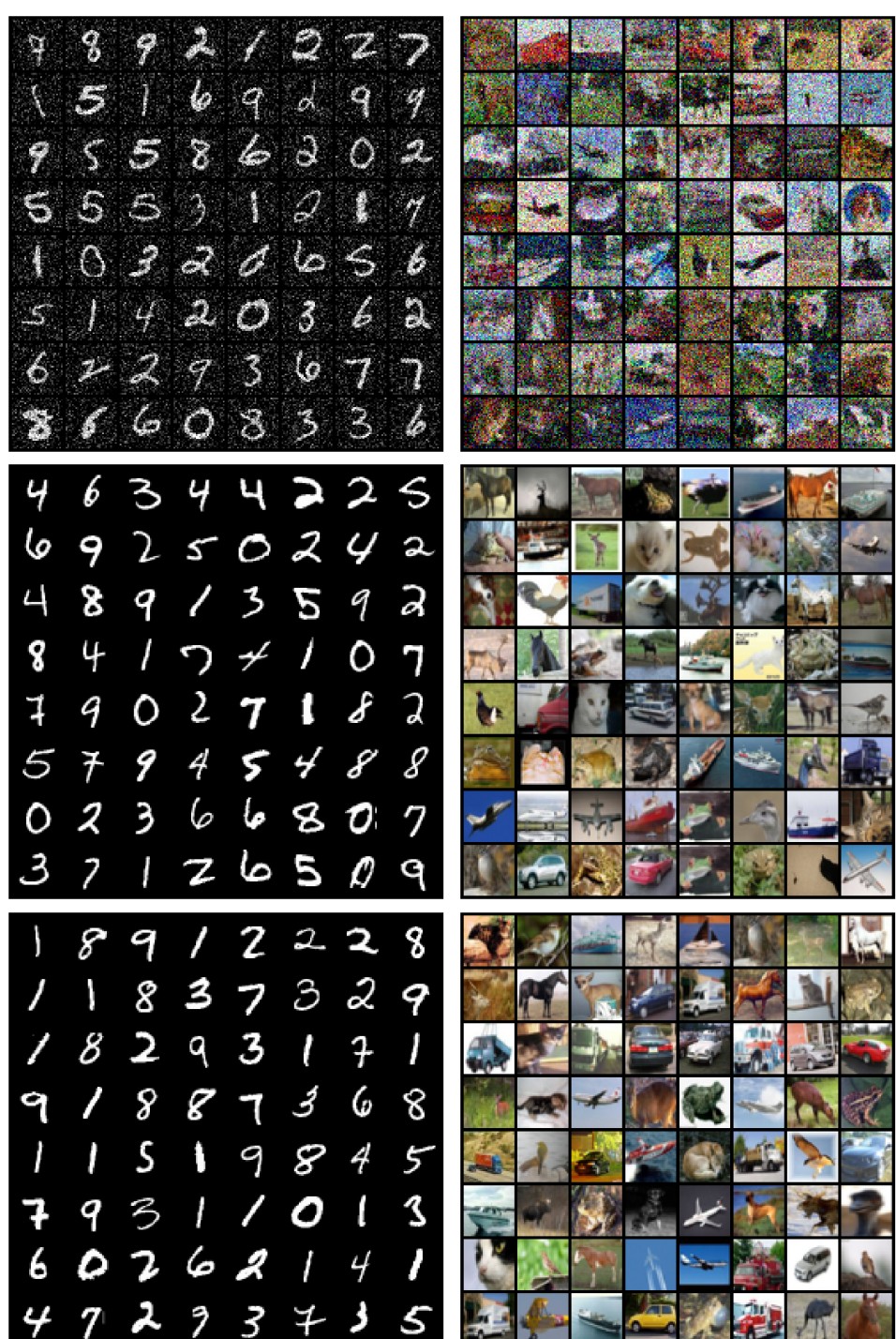

Figure 4: From top to bottom, they are generated by Exact-$x_t$ for $t = 0.8$, D-ODE(rk2) with 8 steps, and Euler-1 with $t = 0.8$.

## B  Proof of Proposition 1

We derive the closed-form expressions for the density $\rho_{\text{EVF}}(t, x)$ and the velocity field $v_{\text{EVF}}(t, x)$. Let $X_t = (1-t)Z + tY$, where $Z \sim f_Z$ and $Y \sim \mathbb{P}_n = \frac{1}{n}\sum_{i=1}^{n} \delta_{y_i}$ are independent.

**Density $\rho_{\text{EVF}}(t, x)$.**  The density of $X_t$ is the convolution of the densities of $(1-t)Z$ and $tY$. The density of $(1-t)Z$ is $\frac{1}{(1-t)^D} f_Z(\frac{\cdot}{1-t})$. The "density" of $tY$ is $\frac{1}{n}\sum_{i=1}^{n} \delta_{ty_i}(\cdot)$. The convolution gives:

$$\rho_{\text{EVF}}(t, x) = \int \frac{1}{(1-t)^D} f_Z\left(\frac{x-z'}{1-t}\right) \left(\frac{1}{n}\sum_{i=1}^{n} \delta_{ty_i}(z')\right) dz'$$

$$= \frac{1}{n}\sum_{i=1}^{n} \frac{1}{(1-t)^D} f_Z\left(\frac{x-ty_i}{1-t}\right).$$

This is the expression for the density of $X_t$, which is a KDE of the scaled data $\{ty_i\}$.

**Velocity Field $v_{\text{EVF}}(t, x)$.**  The velocity field is defined as $v(t, x) = \frac{1}{1-t}(\mathbb{E}[Y|X_t = x] - x)$. We need to compute the conditional expectation $\mathbb{E}[Y|X_t = x]$. Using Bayes' rule for hybrid discrete-continuous distributions:

$$P(Y = y_i | X_t = x) = \frac{f_{X_t|Y}(x|y_i) P(Y = y_i)}{\sum_{j=1}^{n} f_{X_t|Y}(x|y_j) P(Y = y_j)}.$$

Given $Y = y_i$, $X_t = (1-t)Z + ty_i$. The conditional density of $X_t$ is therefore $f_{X_t|Y}(x|y_i) = \frac{1}{(1-t)^D} f_Z(\frac{x-ty_i}{1-t})$. Since $P(Y = y_i) = 1/n$ for all $i$, these terms cancel, and we get:

$$P(Y = y_i | X_t = x) = \frac{\frac{1}{(1-t)^D} f_Z(\frac{x-ty_i}{1-t})}{\sum_{j=1}^{n} \frac{1}{(1-t)^D} f_Z(\frac{x-ty_j}{1-t})} = \frac{f_Z(\frac{x-ty_i}{1-t})}{\sum_{j=1}^{n} f_Z(\frac{x-ty_j}{1-t})}.$$

The conditional expectation is then:

$$\mathbb{E}[Y|X_t = x] = \sum_{i=1}^{n} y_i P(Y = y_i | X_t = x) = \frac{\sum_{i=1}^{n} y_i f_Z(\frac{x-ty_i}{1-t})}{\sum_{j=1}^{n} f_Z(\frac{x-ty_j}{1-t})}.$$

Substituting this into the formula for $v(t, x)$ gives the desired result for $v_{\text{EVF}}(t, x)$.

## C  Proof of Theorem 1 (Projection Effect)

We first state and prove a standard result about nearest-point projections onto smooth manifolds.

**Lemma 1** (Orthogonality of Projection).  Let $\mathcal{M}$ be a $C^1$ manifold in $\mathbb{R}^D$. Let $y \in \mathbb{R}^D$ be a point that admits a unique nearest-point projection $\pi(y) \in \mathcal{M}$. Then the vector $y - \pi(y)$ is orthogonal to the tangent space of $\mathcal{M}$ at $\pi(y)$, denoted $T_{\pi(y)}\mathcal{M}$.

*Proof.* Let $u = \pi(y)$. Let $\gamma : (-\epsilon, \epsilon) \to \mathcal{M}$ be any smooth curve on the manifold passing through $u$ at $t = 0$, i.e., $\gamma(0) = u$. The vector $\gamma'(0)$ is a tangent vector in $T_u\mathcal{M}$. Since $u$ is the point on $\mathcal{M}$ that minimizes the distance to $y$, the function $f(t) = \|y - \gamma(t)\|^2$ must have a minimum at $t = 0$. Its derivative must therefore be zero at $t = 0$:

$$\frac{d}{dt} f(t)\Big|_{t=0} = \frac{d}{dt}\langle y - \gamma(t), y - \gamma(t)\rangle\Big|_{t=0}$$

$$= -2\langle \gamma'(0), y - \gamma(0)\rangle = -2\langle \gamma'(0), y - u\rangle = 0.$$

This holds for any tangent vector $\gamma'(0) \in T_u\mathcal{M}$. Therefore, the vector $y - u$ is orthogonal to the entire tangent space $T_u\mathcal{M}$. □

*Proof of Theorem 1.* Let $\hat{y} = \sum_{i=1}^{n} w_i y_i$ be the generated point, where $w_i > 0$ only if $\|x - y_i\| \le rh$ due to the compact support assumption on $f_Z$. Since $\hat{y}$ is a convex combination of these "active" $y_i$, it must lie within their convex hull. Thus, $\|x - \hat{y}\| \le rh$. By the triangle inequality, for any active training point $y_i$:

$$\|y_i - \hat{y}\| \le \|y_i - x\| + \|x - \hat{y}\| \le rh + rh = 2rh.$$

Let $u_0 = \pi(\hat{y})$ be the projection of $\hat{y}$ onto $\mathcal{M}$. The affine tangent space at $u_0$ is $\mathcal{T}_{u_0} = u_0 + T_{u_0}\mathcal{M}$. By Lemma 1, the vector $\hat{y} - u_0$ is orthogonal to the tangent space $T_{u_0}\mathcal{M}$. This implies that the projection of $\hat{y}$ onto the affine space $\mathcal{T}_{u_0}$ is $u_0$ itself, i.e., $P_{\mathcal{T}_{u_0}}(\hat{y}) = u_0$.

Now, for any active $y_i$, we have $\|u_0 - y_i\| \le \|u_0 - \hat{y}\| + \|\hat{y} - y_i\|$. Because $u_0$ is the closest point on $\mathcal{M}$ to $\hat{y}$, $\|u_0 - \hat{y}\| \le \|y_i - \hat{y}\|$, which gives $\|u_0 - y_i\| \le 2\|\hat{y} - y_i\| \le 4rh$. We can now apply Assumption (ii) of the theorem with $u = u_0$ and $v = y_i$:

$$
\begin{aligned}
\|y_i - P_{\mathcal{T}_{u_0}}(y_i)\| &\le \kappa \|P_{T_{u_0}\mathcal{M}}(y_i - u_0)\|^2 \\
&= \kappa \|P_{T_{u_0}\mathcal{M}}(y_i - \hat{y} + \hat{y} - u_0)\|^2 \\
&= \kappa \|P_{T_{u_0}\mathcal{M}}(y_i - \hat{y})\|^2 \quad \text{(Since } \hat{y} - u_0 \perp T_{u_0}\mathcal{M}) \\
&\le \kappa \|y_i - \hat{y}\|^2 \le \kappa(2rh)^2 = 4\kappa r^2 h^2.
\end{aligned}
$$

Finally, consider the error vector $\hat{y} - \pi(\hat{y})$:

$$
\begin{aligned}
\hat{y} - \pi(\hat{y}) &= \hat{y} - u_0 = \hat{y} - P_{\mathcal{T}_{u_0}}(\hat{y}) \\
&= \sum_{i=1}^{n} w_i y_i - P_{\mathcal{T}_{u_0}}\left(\sum_{i=1}^{n} w_i y_i\right) \\
&= \sum_{i=1}^{n} w_i (y_i - P_{\mathcal{T}_{u_0}}(y_i)) \quad \text{(by linearity of projection onto an affine subspace).}
\end{aligned}
$$

Taking the norm and using the triangle inequality and the fact that $\sum w_i = 1$:

$$
\begin{aligned}
\|\hat{y} - \pi(\hat{y})\| &= \left\| \sum_{i=1}^{n} w_i (y_i - P_{\mathcal{T}_{u_0}}(y_i)) \right\| \\
&\le \sum_{i=1}^{n} w_i \|y_i - P_{\mathcal{T}_{u_0}}(y_i)\| \\
&\le \max_{i:w_i>0} \|y_i - P_{\mathcal{T}_{u_0}}(y_i)\| \le 4\kappa r^2 h^2.
\end{aligned}
$$

$\square$

# D  PROOF OF THEOREM 2 (DIVERSITY)

Consider arbitrary open neighborhood around $u$ and call it $N(u)$. Define $\phi(x) = \pi(y(x))$. Since $\phi$ is a continuous function, $\phi^{-1}(N(u))$ is an open set in $\mathbb{R}^D$. Under Assumption (ii), $x$ belongs to this set. Thus it is an open neighborhood of $x$. Under Assumption (i), any neighborhood of $x$ has positive probability mass, so $\phi^{-1}(N(u))$ has positive probability mass. This implies for every open neighborhood of $u$, the probability mass is positive.

# E  HIGH-DIMENSIONAL MANIFOLD VISUALIZATIONS

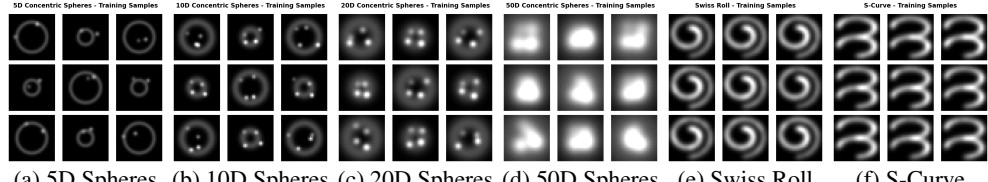

(a) 5D Spheres    (b) 10D Spheres    (c) 20D Spheres    (d) 50D Spheres    (e) Swiss Roll    (f) S-Curve

Figure 5: Sample visualizations of high-dimensional manifold datasets projected to 2D pixel grids. Each image shows training samples from different manifold structures: concentric spheres in various dimensions (5D, 10D, 20D, 50D), Swiss roll, and S-curve manifolds.

# F  2D SYNTHETIC DATASET VISUALIZATIONS

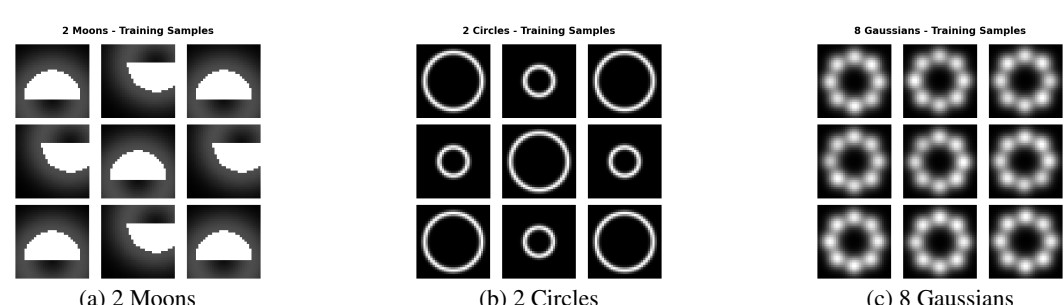

(a) 2 Moons      (b) 2 Circles      (c) 8 Gaussians

Figure 6: Sample visualizations of 2D synthetic datasets converted to pixel grids. These datasets represent classic machine learning benchmarks: two moons, two circles, and eight Gaussians distributions.

## G   HIGH-DIMENSIONAL MANIFOLD PR RESULTS

## H   2D SYNTHETIC DATASET PR RESULTS

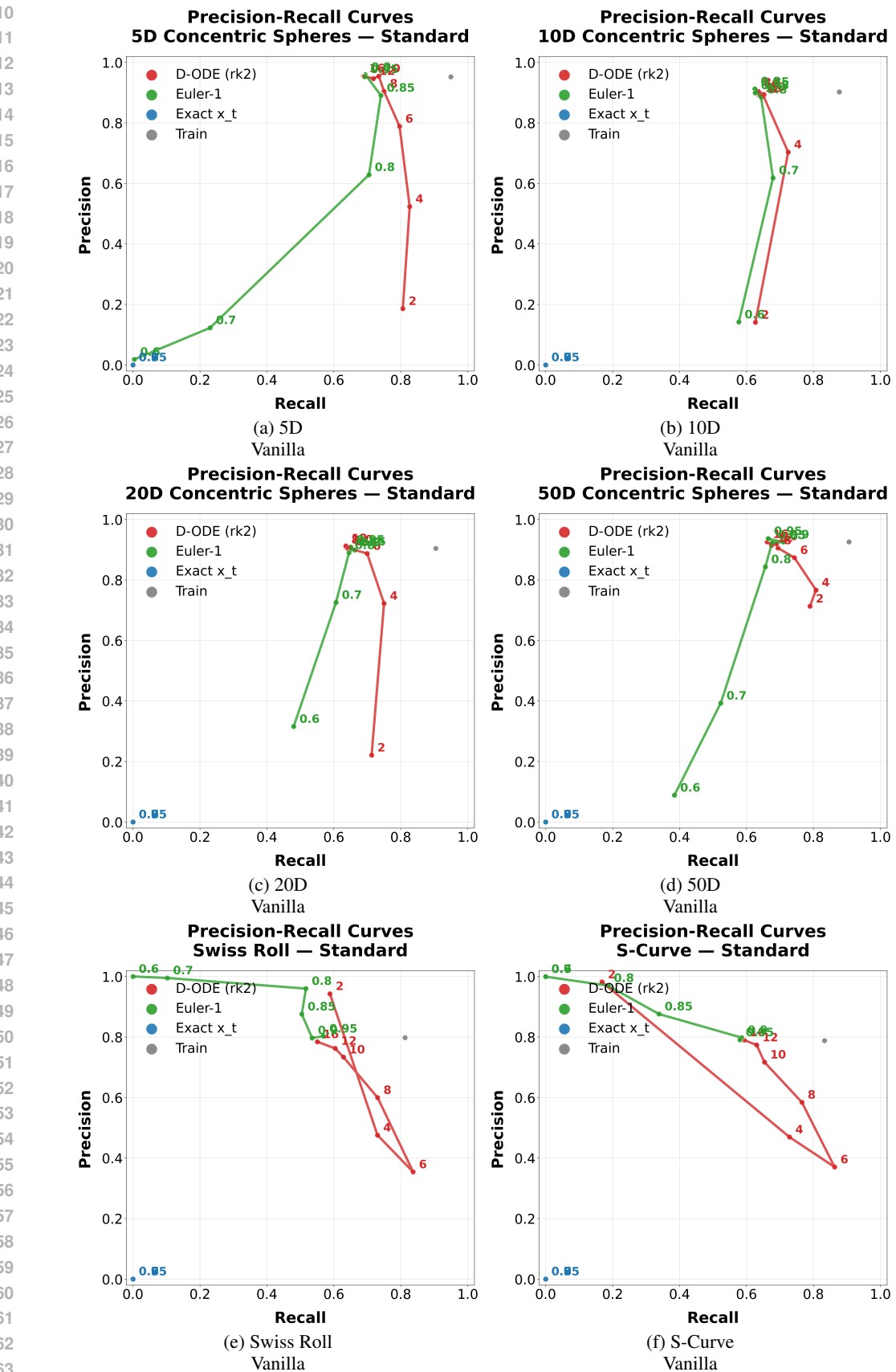

Figure 7: Precision-Recall curves for high-dimensional manifold datasets using vanilla EVF evaluation (no novelty filtering). Results show EVF performance across different manifold complexities and dimensionalities.

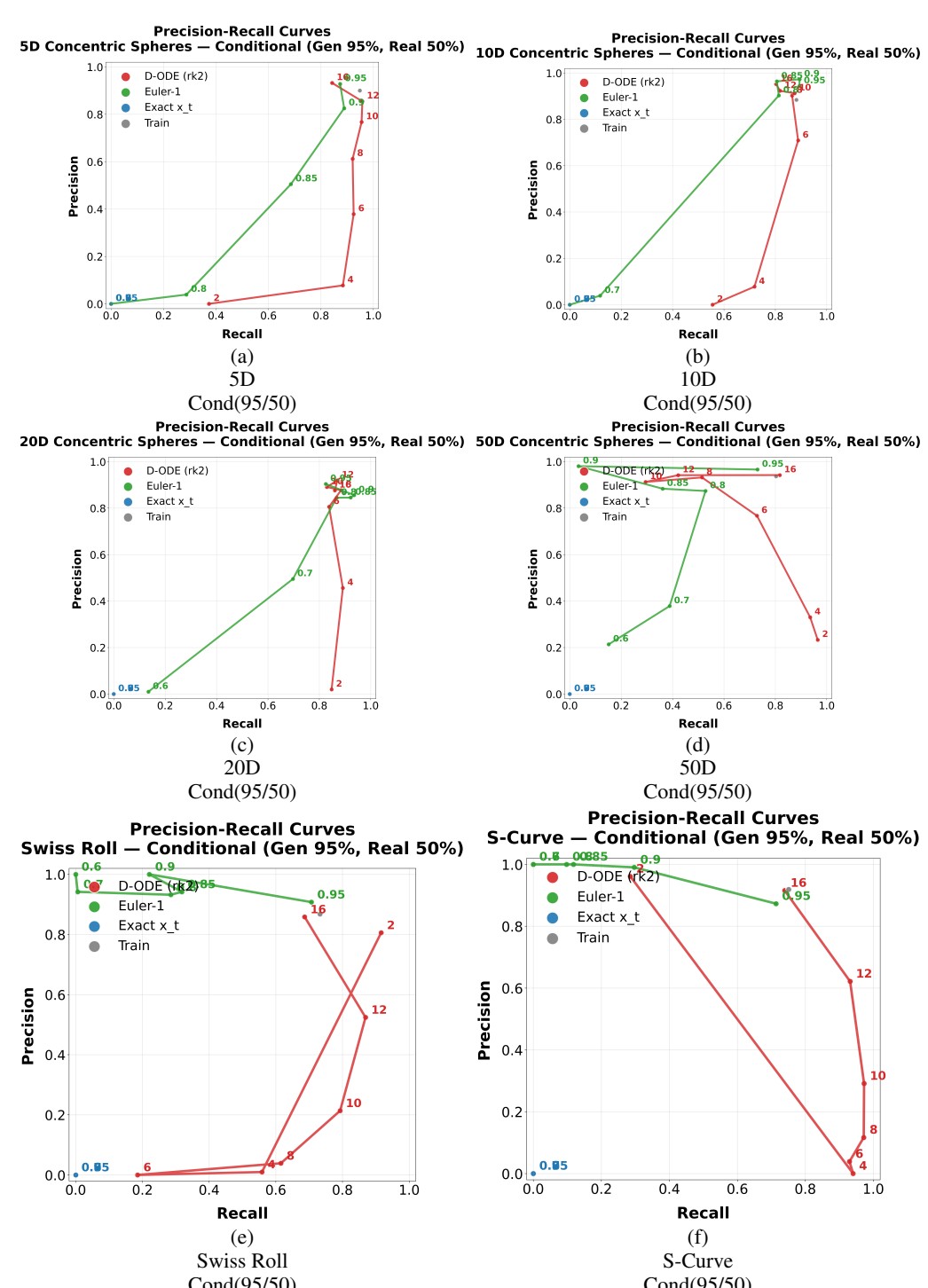

Figure 8: Precision-Recall curves for high-dimensional manifold datasets using conditional EVF evaluation with novelty filtering (95th percentile for generated samples, 50th percentile for real samples). This filtering focuses evaluation on novel, high-quality generated samples.

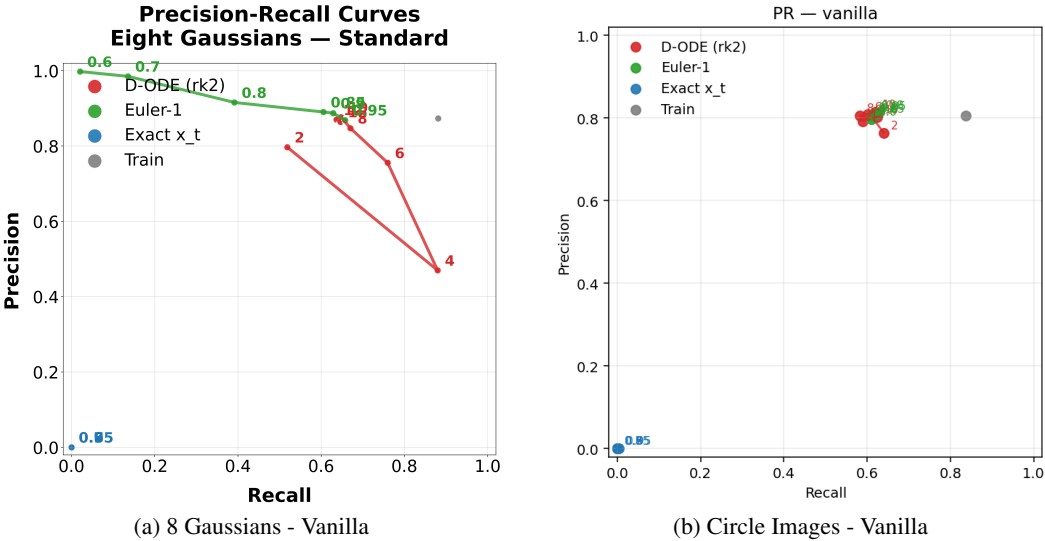

(a) 8 Gaussians - Vanilla

(b) Circle Images - Vanilla

Figure 9: Precision-Recall curves for 2D synthetic datasets using vanilla EVF evaluation. Results demonstrate EVF performance on classic machine learning benchmarks converted to pixel space.

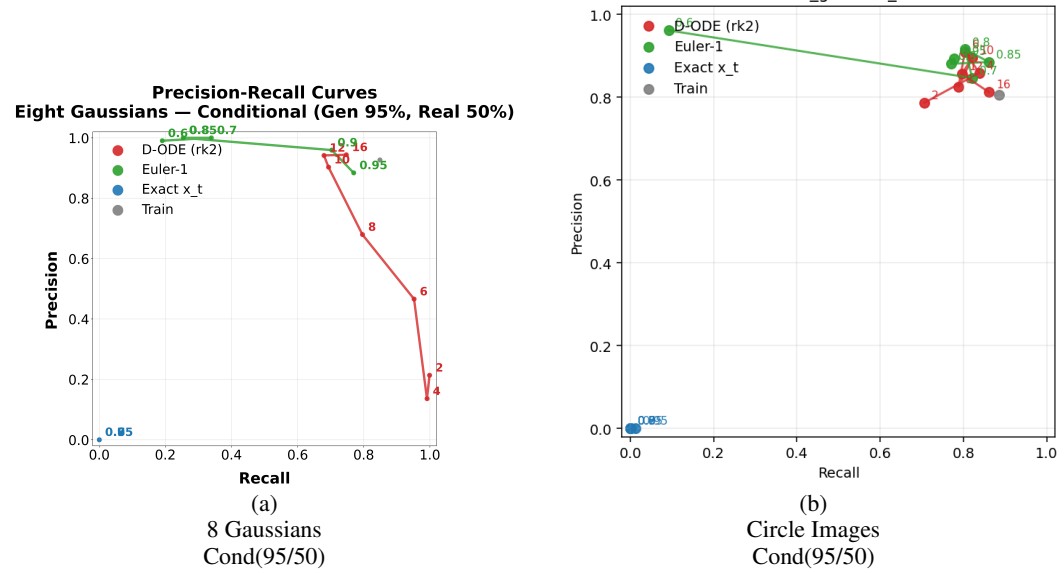

(a)
8 Gaussians
Cond(95/50)

(b)
Circle Images
Cond(95/50)

Figure 10: Precision-Recall curves for 2D synthetic datasets using conditional EVF evaluation with novelty filtering (95/50 thresholds). The conditional evaluation provides more discriminative assessment by focusing on novel generated samples.

## I  REAL IMAGE DATASET PR RESULTS

Enlarged versions of the PR plots for real images.

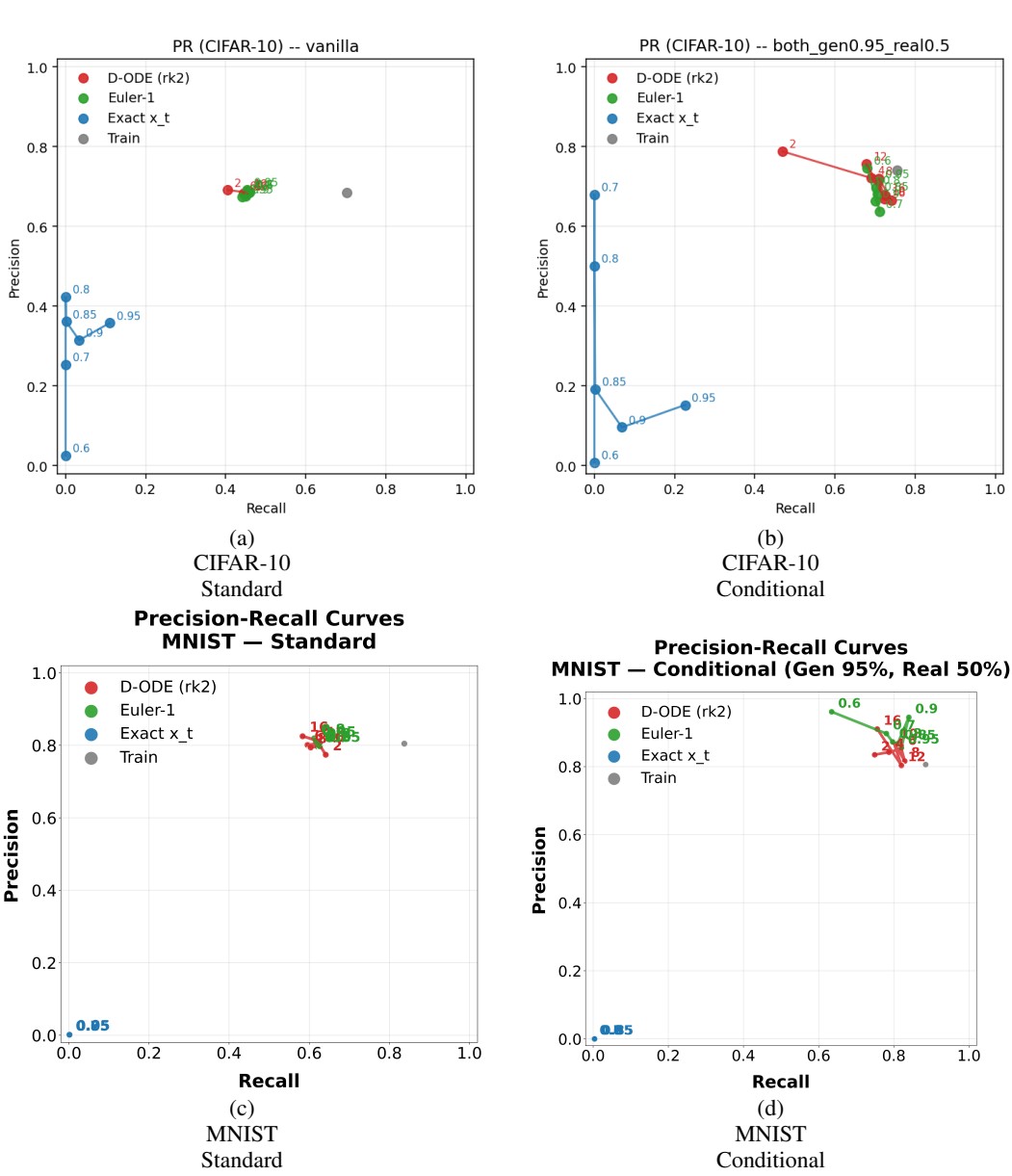

Figure 11: Precision-Recall curves for real image datasets comparing standard EVF evaluation and conditional evaluation with novelty filtering (95th percentile for generated samples, 50th percentile for real samples). Results show EVF performance on (a,b) CIFAR-10 and (c,d) MNIST datasets.

## J  LARGE LANGUAGE MODEL (LLM) USAGE

We used LLMs (GPT-5 and Gemini) to help with writing code and polishing the writing of the paper.

