# OpenReview forum: "Flow Matching Generalizes Through Discretization Bias"
_ICLR.cc/2026/Conference — Submitted to ICLR 2026_

### Official Review · Reviewer_kPh3 · 2025-10-27

**Soundness:** 1
**Presentation:** 2
**Contribution:** 1
**Rating:** 0
**Confidence:** 4

**Summary:**

This paper studies the generalization of flow matching models and claims that the generalization ability lies in the discretization bias and proposes a method that enables generalization through integrating the empirical optimal velocity field with some deliberate discretization bias/error.

**Strengths:**

The paper studies the generalization of flow matching models, which is an important problem.

**Weaknesses:**

1. Insufficient related work discussion. For example, modifying the closed form velocity for generalization is already explored in [1].
2. Theorem 1 is restrictive and lacks justification of a key assumption. Specifically, it assumes the density function $f_Z$ has compact support, which is not the case for the diffusion model, where $f_Z$ is the density of the Gaussian distribution and the support is $\mathbb{R}^d$. Additionally, it makes a simplification by assuming $t$ is close to 1 and fails to justify whether the obtained rate is still valid in the actual setting. Most importantly, the result obtained in Theorem 1 is relatively straightforward, and the quadratic dependency in h comes from the quadratic bound in assumption (ii), which is never discussed nor justified.
3. It was not clear from the paper whether the obtained sampling on Cifar-10 was novel generation or memorized training data. Especially, note that in Figure 4, there are repeated images in the Cifar-10 visualization (two identical frogs), so despite the statistical analysis in the paper, whether the proposed method truly generalizes is still unclear. I ran the provided code and conducted a nearest-neighbor search on the training data, and found that the samples are just memorized training data. Therefore, the core claim that discretization bias leads to novel generation is not supported by the experiments.


[1] Scarvelis, Christopher, Haitz Sáez de Ocáriz Borde, and Justin Solomon. "Closed-form diffusion models." arXiv preprint arXiv:2310.12395 (2023).

**Questions:**

1. Whether the obtained sampling on Cifar-10 was novel generation or memorized training data.
2. Can you provide the FID on CIFAR-10, ImageNet, and the corresponding visualization of generated samples?
3. What is the run-time complexity?

---

### Official Review · Reviewer_nRry · 2025-10-28

**Soundness:** 2
**Presentation:** 2
**Contribution:** 3
**Rating:** 4
**Confidence:** 3

**Summary:**

The paper proposes a non-parametric diagnostic framework, Estimated Vector Field (EVF), for analyzing flow matching through the lens of kernel density estimation. With EVF, the authors theoretically show that errors introduced by numerical discretization can promote generalization by implicitly projecting generated samples toward the true data manifold, whereas samples obtained by integrating the exact ODE are identical as train data. To validate this claim, the authors introduce NcPR as evaluation metric, designed to separate genuine distributional generalization from sample memorization.

**Strengths:**

1. Introduces a non-parametric framework (EVF) for analyzing flow matching by modeling the target distribution as the empirical data distribution, enabling direct examination of how the learned vector field behaves on the training set.

2. Provides a theoretically supported interpretation of discretized ODE integration, showing that discretization induces a projection effect toward the data manifold

3. Proposes NcPR, an evaluation metric for generative models that explicitly controls for training-set memorization, allowing more reliable measurement of precision and recall.

4. The paper presents strong internal ablation result with various datasets, demonstrating the effectiveness and robustness of the proposed method through experiments.

**Weaknesses:**

1. The theoretical analysis in the paper is limited to the Euler-1 method. For $\textbf{D-ODE}$, whose discretization involves more than two steps, the effect of accumulation of error is not directly addressed. Consequently, this work does not provide a guarantee that $\textbf{D-ODE}$ is not worse than $\textbf{Euler-1}$.

2. In the experimental section, there is insufficient theoretical analysis and discussion of how the EVF results vary as a function of the step size $s$. This lack of analysis makes it difficult to assess the robustness and practical implications of the method.

3. This work does not include experiments with various solvers. As a result, it is unclear whether the proposed theoretical assumptions and analyses hold for other solvers under discretized.

**Questions:**

1. Are the $\textbf{Exact-$x_t$}$ results explicitly dependent on $t$? I am particularly interested in whether there exists a discretization sweet spot that optimally balances the generalization gap.

2. Between $\textbf{Euler-1}$ and $\textbf{D-ODE}$, which method is better in terms of theoretical aspects? In other words, does the hybrid approach $-$ where the trajectory is integrated continuously with an exact ODE solver and only the final step is discretized $-$ perform better than a fully discretized integration method, or vice versa?

3. What is the exact training configuration of $\textbf{Train}$? Was the model trained solely using the standard Flow Matching objective?

4. The theoretical foundation presented in this paper is sound, but I would like to check if these theoretical findings are effectively applied in real generative models. How does the model's performance in terms of NcPR measure up when using discretized or adaptive solvers (e.g., dopri5), in a real model trained with flow matching objectives?

---

### Official Review · Reviewer_m5BT · 2025-10-31

**Soundness:** 3
**Presentation:** 3
**Contribution:** 2
**Rating:** 4
**Confidence:** 4

**Summary:**

The paper studies why Flow Matching (FM) samplers can “generalize” by isolating the role of time discretization. It replaces the learned velocity $v_\theta$ with a closed-form Empirical Velocity Field (EVF) built from the empirical data distribution, showing that the continuous-time EVF flow yields a KDE that collapses to the training set as $t!\to!1$ (non-generative), while a discretized step near $t!=!1$ becomes a Nadaraya–Watson–type weighted average that pulls states toward the data manifold (“projection effect”). The authors prove local error/coverage properties and present visual experiments suggesting that discretization bias—rather than the network—can create on-manifold samples “between” training points.

**Strengths:**

Originality: Clear, orthogonal lens on FM—fix the velocity (EVF) to isolate discretization bias from network approximation, which yields fairly striking conclusions.
Quality (theory): Clean derivations; the projection-like characterization (one-step ≈ kernel regression) is insightful and links FM sampling to classical nonparametrics.
Clarity: Core constructions (EVF density/velocity, one-step update) are easy to follow; figures help intuition.
Significance (potential): If robust, the results inform solver design (step placement near $t!\approx!1$), few-step generation, and training objectives that align with the projection effect.

**Weaknesses:**

Scope/claims: The “perfectly interpolating gaps” phrasing overstates what is proven; theory supports on-manifold attraction and positive density, not global gap-filling. The figures themselves show residual under-coverage.
High-dimensional realism: Empirics focus on low-D/toy setups; it remains unclear how EVF (which sums over all samples) scales to images/audio or how approximate neighbors/inducing sets affect guarantees.
Separation assumption: The paper treats discretization and network biases as orthogonal; in practice they interact. No experiments quantify this interaction under a learned $v_\theta$.
Comparative context: Limited discussion vs. related paradigms (e.g., Rectified Flow, Consistency Models, denoising/score projection). A head-to-head showing what EVF-discretization alone achieves vs. a single denoise/consistency step would sharpen novelty.

**Questions:**

Can the discretization be viewed as a form of noise injection on discrete data? If yes, please formalize the equivalence (or non-equivalence) between the near-final EVF update and adding noise at $t!=!1$?

---

### Meta-Review · Area_Chair_CvVV · 2026-01-06

**Summary:**

No rebuttal for the reviewers' comments.  Reviewers are negative. Reject.

**Reviewer Concerns:**

.

**Reviewer Scores:**

.

---

### Decision · Program_Chairs · 2026-01-26

Reject